# A theoretical framework to improve the adoption of green Integrated Pest Management tactics
Peng Han [1,2] ✉, Cesar Rodriguez-Saona [3] ✉, Myron P. Zalucki [4,5], Shu-sheng Liu[6] & Nicolas Desneux[7]

Sustainable agriculture relies on implementing effective, eco-friendly crop protection strategies. However, the adoption of these green tactics by growers is limited by their high costs resulting from the insufficient integration of various components of Integrated Pest Management (IPM). In response, we propose a framework within IPM termed Multi-Dimensional Management of Multiple Pests (3MP). Within this framework, a spatial dimension considers the interactive effects of soil-crop-pest-natural enemy networks on pest prevalence, while a time dimension addresses pest interactions over the crop season. The 3MP framework aims to bolster the adoption of green IPM tactics, thereby extending environmental benefits beyond crop protection.

Modern agriculture has long been seeking effective and environmentally sound strategies to help growers protect crops against pests, which otherwise can result in substantial revenue losses. "Pests" are "any species, strain or biotype of plant, animal or pathogenic agent injurious to plants or plant products," as defined by the International Plant Protection Convention[1]. In this context, authors mainly refer to pests as insect pests and plant pathogens. The goal is to maintain these pests at levels below economical injury, employing diverse methods with minimal health and environmental risks[2]. To achieve this goal, a range of 'green' (i.e., environmentally and human-health benign) management tactics are now available. These methods include biological control using arthropods and microbials[3,4], biotechnology using resistant cultivars (such as a plant variety produced by selective breeding or genetic engineering)[5,6], physicochemical control based on sensory cues (e.g., use of insect- and plant-derived volatile compounds)[7], and agronomic methods (e.g., soil management)[8]. These tactics exhibit various advantages and drawbacks contingent on factors such as application context, adoption cost, and biological characteristics of the pest. For instance, the efficiency of biological control by releasing arthropod natural enemies is notable in greenhouses[9], but more complex in open fields[10]. Insect sex pheromone-based mating disruption is effective in reducing pest population density, yet it poses a significant cost for growers[11]. Plant resistance breeding, while often tailored to target a specific damaging pest, may inadvertently

lead to the development of insect resistance and the emergence of new pest challenges[12]. Frequently, these green tactics find adoption within the framework of Integrated Pest Management (IPM), a long-standing and highly supported paradigm[13].

However, despite their presence in publications and textbooks, green tactics are largely underutilized in the practical implementation of IPM in fields[14]. Apart from socioeconomic factors, the limited adoption is primarily attributed to high costs and suboptimal performance when growers employ multiple tactics simultaneously without considering their synergy and coverage for managing multiple pests within a single crop system—a concept known as the Simple Mixed Inputs (SMI) approach. This scenario often arises when growers are facing challenges from several pests affecting one crop. In practice, many of the IPM green tactics fail to progress to the implementation stage in fields due to low cost-effectiveness, primarily stemming from a lack of knowledge on interactions between or among these tactics not only by growers but also by outreach specialists and researchers. Given the specificity of expertise, researchers, mainly entomologists and plant pathologists, tend to focus more on basic research aimed at developing and optimizing individual plant protection approaches or addressing a single pest. Consequently, they pay less attention to their integration, which are more applied in nature. From a philosophical standpoint, the former relies on reductionism as a way of thinking, while the latter leans more

[1]Institute of Biodiversity, School of Ecology and Environmental Sciences, Yunnan University, Kunming 650500, China. [2]Southwest United Graduate School (SWUGS), Kunming 650092, China. [3]Department of Entomology, Rutgers University P.E. Marucci Center, Chatsworth, NJ, USA. [4]School of the Environment, The University of Queensland, Brisbane, QLD 4072, Australia. [5]Shandong Engineering Research Center for Environment-Friendly Agricultural Pest Management, College of Plant Health and Medicine, Qingdao Agricultural University, Qingdao 266109, China. [6]Institute of Insect Sciences, College of Agriculture and Bio-technology, Zhejiang University, Hangzhou 310058, China. [7]Université Cote d'Azur, INRAE, CNRS, UMR ISA, 06000 Nice, France. ✉e-mail: penghan@ynu.edu.cn; crodriguez@njaes.rutgers.edu

towards holism and/or systematology. Clearly, there is a need for a holistic science of IPM that emphasizes systematic studies on the compatibility and optimization of concurrently implemented actions associated with at least two pest management tactics[15].

To overcome the limitations of the SMI approach, it is imperative to address two key bottlenecks. The first bottleneck involves low synergy, characterized by inadequate integration of multiple tactics in managing a specific pest. The interactions between two tactics against a single pest can be synergistic, additive, or antagonistic. For instance, plant physical defenses like trichomes, which can deter pest oviposition and locomotion, may inadvertently disrupt the natural enemies of the pest[16]. Crop breeders are expected to play a role in developing cultivars that are suitable for specific biocontrol agents[17]. Additionally, the heavy application of nitrogen is known to compromise the efficacy of biocontrol agents through bottom-up effects[18]. However, the bottom-up effects of nitrogen inputs on biocontrol agents are not consistently evident, as observed in other case studies[19,20].

The second bottleneck is low coverage, wherein management is predominantly centered on a single pest rather than on a specific crop that may be affected by several pests. In other words, IPM is proposed to be crop-centered rather than pest-centered. Often, a crop is simultaneously or sequentially infested or infected by multiple insect pests and pathogens, resulting in complex interactions[14]. It is crucial to recognize that an excessive focus on the dominant pest may trigger shifts in the pest assemblage, leading to a heightened prevalence of secondary pests. For example, the widespread

adoption of genetically modified (GM) *Bacillus thuringiensis* (*Bt*) cotton has effectively managed the cotton bollworm but it has also been implicated in the appearance and subsequent spread of non-target pests at the agro-landscape level[12].

A new theoretical framework is essential to overcome these bottlenecks, aiming to deepen our comprehension of the intricate interactions among various green tactics and boost their adoption within the IPM paradigm. It is crucial to note that our intention is not to rebrand IPM, but rather to provide a guiding framework for enhancing the adoption of green IPM tactics. This framework seeks to serve as a catalyst for refining and expanding the application of environmentally sustainable practices within the existing IPM framework.

## New theoretical framework

Recent progress in understanding tri-trophic interactions (involving plants, herbivores, and natural enemies), bottom-up versus top-down forces[5], indirect interactions among organisms[21,22], and plant-soil feedback[23] have paved the way for a more nuanced and engineered approach to pest management. In light of these insights, we propose a novel theoretical framework termed Multi-Dimensional Management of Multiple Pests (3MP). The philosophy underpinning 3MP is to strategically design both above- and below-ground ecological elements to synergistically control multiple harmful organisms in a cropping system throughout the entire growing season (Fig. 1). The primary objective is to encourage researchers to leverage

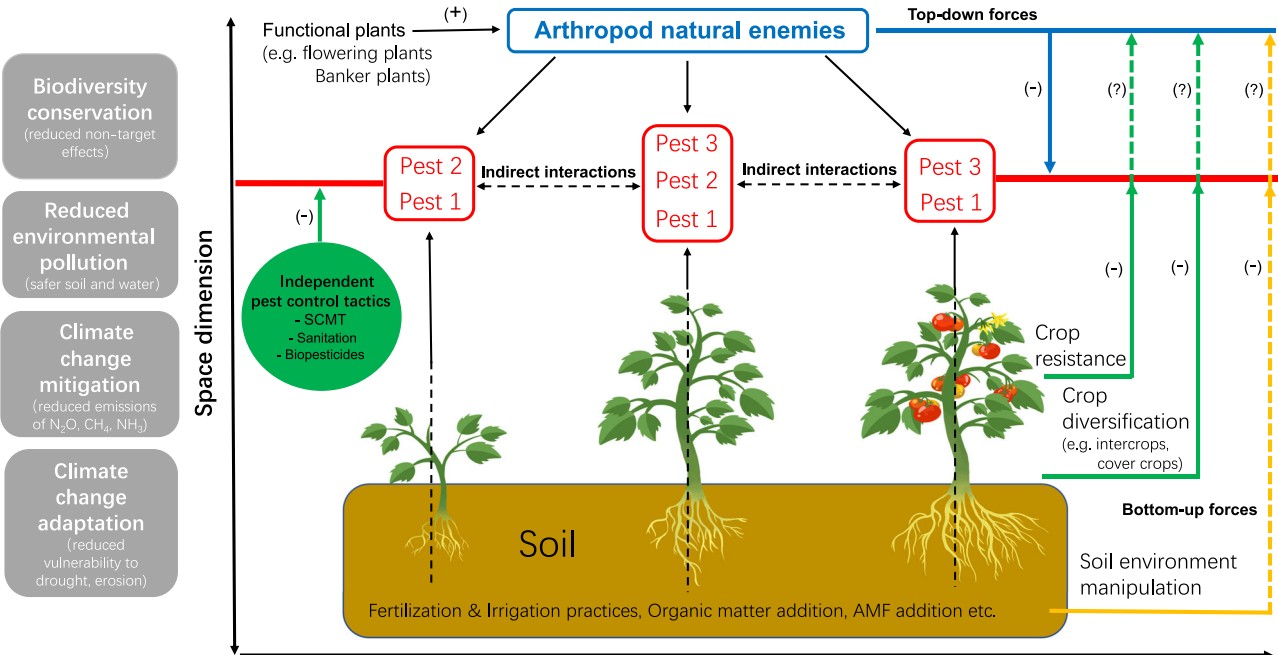

**Fig. 1 | A new theoretical framework–Multi-Dimensional Management of Multiple Pests (3MP)–enabling a nuanced and holistic approach to the management of multiple pests across the cropping season.** The first dimension of the framework is "space". Soil environment manipulation, crop resistance (constitutive and/or induced), and crop diversification are engineered to provide bottom-up forces, influencing not only the second trophic level but also reaching the third trophic level and influencing top-down forces. Natural enemy release and conservation through functional plants contribute to top-down forces. The combined action of both forces works synergistically to lower pest prevalence. The second dimension of the framework is "time". Pest 1 could be a plant pathogen persisting across the season, while Pest 2 and 3 could be insect pests appearing during the early and late seasons, respectively, and potentially serving as disease vectors. Indirect interactions among these pests, or between pests and pathogens, may impact the sign and magnitude of bottom-up and top-down forces on pest prevalence. This allows for a more precise manipulation of these ecological drivers to enhance the synergy and coverage of pest management in a given crop system. Arthropod natural enemies (predators and

parasitoids) could target a wide range of insect pest species throughout the season. Functional plants, often non-crop plants, provide shelter and/or non-prey supplementary food to nourish arthropod natural enemies, such as flowering plants and banker plants. The framework also incorporates arbuscular mycorrhizal fungi (AMF) and addresses greenhouse gas (GHG) emissions. Other independent pest control tactics include sanitation recommendations (timely removal of infested plants/plant organs), sensory cue-based mass trapping (SCMT) using lure-and-kill methods based on sex pheromones, artificial diet, and light, as well as mating disruption. Both bottom-up and top-down forces can manifest as direct or indirect effects, represented by solid and dashed lines, respectively. Research endeavors within this framework yield additional environmental benefits, including biodiversity conservation in agroecosystems, reduced environmental pollution, and climate change mitigation and adaptation in agro-ecosystems. This holistic approach aligns with sustainable agriculture practices and contributes to the overall well-being of ecosystems and agricultural landscapes.

**Fig. 2 | Guidelines in the IPM and 3MP framework.** The 3MP theoretical framework falls within the paradigm of IPM.

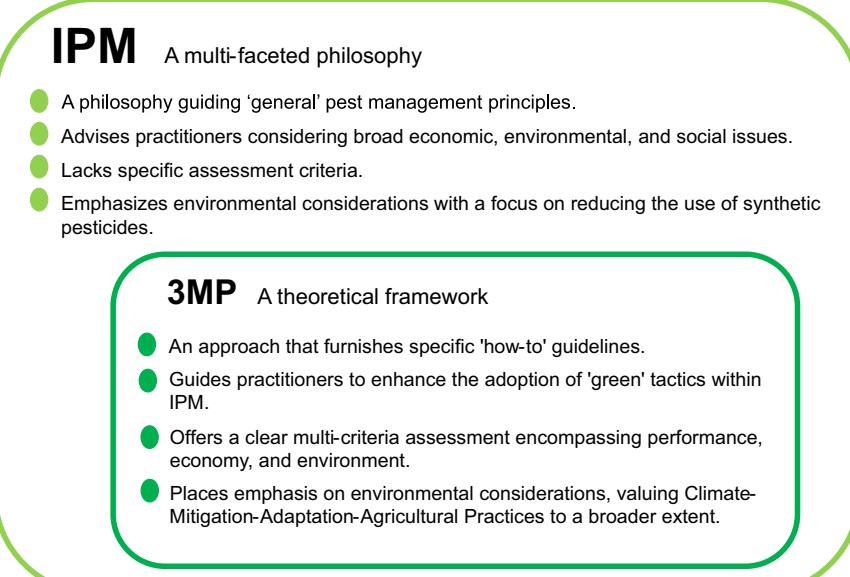

this conceptual framework in identifying complex interactions among various management options within the soil-crop-pests-natural enemies multi-trophic networks over time. Subsequently, this knowledge can be transferred to outreach specialists to aid in the design of improved IPM packages. The ultimate goal is to promote higher adoption of green pest control tools and techniques among growers, leading to the development and implementation of more environmentally sustainable IPM packages (i.e., greener IPM packages). Figure 2 provides guidelines under the IPM and 3MP frameworks.

## Improving synergy

Mounting evidence suggests that key IPM tactics, including biological and behavioral control, constitutive or induced crop resistance, and soil trait manipulation, could be more effectively integrated for pest management[15,24]. An example of this integration is seen in some insect pests feeding on insect-resistant crops (e.g., Bt crops). These pests may develop and survive poorly (bottom-up effects), making them more susceptible to predation by natural enemies (top-down effects). Consequently, insect-resistant crops and biological control may act synergistically to manage pests within IPM programs[25]. Our 3MP framework proposes looking at interactions between bottom-up and top-down effects, resulting in the synergistic suppression of insect pests and thus mitigating the low synergy bottleneck. Further research is imperative to explore synergistic interactions among pest species, microbial controls, and even insecticides. Determining synergy between different tactics can be achieved through full-factorial manipulative experiments. Rather than employing a snapshot approach, researchers should collect trajectory data, utilizing spatial and temporal replication, to better understand which tactics act synergistically. Additionally, cultivating functional plants that support biological control (e.g., flower strips, banker plants, and natural enemy source habitats) could enhance top-down pest control by nourishing generalist predators and sustaining their populations when the focal prey is scarce[26,27]. Moreover, alternative independent pest control tactics (e.g., lure-and-kill) can offer additional suppression of pest populations without disrupting established bottom-up and top-down control strategies.

## Improving coverage

Another critical issue involves the coverage of pest management packages, where many IPM strategies tend to concentrate on a single pest species rather than addressing the entire pest assemblage. However, across the cropping season, interactions among multiple pests and diseases significantly shape pest dynamics, influencing prevalence and crop damage.

Indirect interactions, often plant-mediated, play a crucial role. Research indicates that early-season herbivores can induce phenotypic changes in host plants that affect later-season pests[21]. For instance, plant-mediated tripartite interactions can occur among a sap-feeding insect, a leaf-chewing insect, and a fungal pathogen due to shared phytohormonal pathways[28]. Theoretically, plant-mediated negative indirect interactions could be harnessed to manage a more damaging pest by sustaining a small population of a less harmful species[21]. Interactions can also be mediated by shared natural enemies. Generalist predators play a crucial role in reliable conservation biological control. Indirect interactions between prey species sharing a common generalist predator can influence both community dynamics and the efficacy of biological control[22]. For instance, alternative prey foods can either benefit or hinder focal prey suppression by a shared predator, depending on the degree of prey phenological synchrony[29]. It is likely that both plant- and natural enemy-mediated indirect interactions operate simultaneously. Despite a growing body of literature highlighting these indirect interactions[21,22], they are often overlooked in crop protection practices. Therefore, examining indirect interactions closely, considering the specific phenology of pests in each crop, is essential to achieve a holistic and sustainable management approach for multiple pests and to overcome the low coverage bottleneck.

## A holistic approach: could it help us design greener IPM packages?

In agricultural crops, it is crucial to recognize that space and time dimensions are interdependent, and their integration is essential to achieve the 3MP framework. However, few studies have taken this integrated approach thus far. For instance, in *Brassica oleracea* L., the type of fertilizer demonstrated bottom-up effects on the outcomes of indirect interactions between a phloem feeder and a leaf chewer[30]. In a four-species diamond-shaped food web, plant nutrient inputs exhibited bottom-up effects on plant- and natural enemy-mediated indirect interactions between a leaf miner and an aphid, resulting in the holistic suppression of both pest populations through increased predation and enhanced plant resistance[31]. Understanding how multiple driving forces interact through bottom-up and top-down effects, and how they influence the sign and magnitude of indirect interactions among insect pests, remains an unexplored area. Additionally, while studies with a multi-scale hierarchical design are essential[32], it is unclear how functional processes, grounded in bottom-up, top-down forces, and indirect interactions, are linked to ecological consequences for pest populations over an extended spatial scale from field to landscape. The 3MP theoretical framework provides an unique opportunity for such endeavors. Conducting

factorial manipulative experiments with high replicates in the field, although labor-intensive, can unravel the dynamics of pest prevalence and disentangle cause-and-effect relationships. Chemical and molecular analyses of plant, insect herbivore, pathogen, and natural enemy samples are essential for revealing the mechanisms underlying the observed relationships. Ultimately, quantifying crop yield and quality is necessary for a comprehensive cost-benefit analysis. The information gleaned from specific experiments on a given crop will guide outreach specialists in deciding which green tactics could (or could not) be included in an IPM package, to what extent they should be adopted, and the timing of their adoption throughout the season. This framework facilitates the design of an IPM package with a high level of integration, considering factors such as technology readiness level, market availability, ease of use, cost of relevant techniques, and subjective acceptance by growers.

## Environmental considerations
While economic benefits often take precedence for growers, it is crucial to note that IPM packages designed based on the 3MP framework are assumed to not only enhance economic gains but also reduce contamination of produce, soil, and groundwater, lower greenhouse gas emissions, and contribute to biodiversity conservation in agroecosystems. This emphasis on bottom-up forces modulated by agricultural practices, coined as Climate-Mitigation-Adaptation-Agricultural Practices (CMAAPs), aligns with the idea proposed by Murrell[33]. For instance, practices such as drip fertigation in cropping systems have the potential to reduce nitrous oxide ($N_2O$) emissions, providing climate mitigation opportunities[34]. Additionally, these practices are assumed to modulate crop-pest-natural enemy multitrophic interactions through bottom-up effects, which may lower pest prevalence. However, specific research on the potential benefits of these systems for pest management is still in its early stages. Arable production systems contribute significantly to greenhouse gas emissions, and in the context of global agreements like the Paris Agreement and the Sharm el-Sheikh Implementation Plan, more research is essential to identify agricultural practices that mitigate climate change impacts and improve pest management[33]. For example, intercropping has been shown to increase water use efficiency, protect soil from extreme climate events, reduce greenhouse gas emissions, and enhance crop yields through improved pest control[35]. Similarly, cover crops integrated into crop rotations have high potential for climate change mitigation and adaptation[36], although their impacts on crop resistance to pests and biological pest control are not extensively measured. These areas should be actively explored in future research within the 3MP framework. Assessing the potential role of CMAAPs in IPM could contribute to the development of climate-smart agriculture, which aims to ensure secured productivity, increase the adaptation of agricultural systems to climate change, and enhance the capacity of climate change mitigation[37].

## Criteria for assessing the increased adoption of 'green' tactics in IPM
A Performance-Economy-Environment (PEE) multi-criteria assessment could be employed to examine the increased adoption of green tactics in IPM. During field experiments utilizing the 3MP framework, a set of indicators should be assessed for various combinations of individual tactics assumed to compose the IPM packages. The P component in the PEE multi-criteria assessment involves evaluating pest management performance, focusing on indicators such as pest prevalence and damage levels. Quantifying crop yield losses is essential, considering the economic aspect as well. The first E aspect encompasses an assessment of the economic cost and return of different combinations of management tactics, calculating all resource inputs and market returns for growers (i.e., cost-effectiveness). The second E considers the environmental and sustainability aspects. This includes evaluating reduced environmental pollution, particularly in soil and water, resulting from decreased inputs of agro-chemicals (e.g., synthetic pesticides, fertilizers). Climate change

mitigation is also considered, particularly when precision fertilization methods like drip systems are adopted. Additionally, climate change adaptation is evaluated, particularly in terms of reduced vulnerability to drought and erosion when cover crops are grown, among other factors. The PEE multi-criteria assessment is not only applied to the elaborated IPM package based on 3MP but it also extends to the SMI approach and the sample-spray-and pray approach[38]. Beyond the PEE aspects, the 3MP framework is designed to uphold ecological well-being. For example, it could offer protection to pollinators from the harmful effects of pesticides, aligning with the goals of Integrated Pest and Pollinator Management (IPPM) as an expanded framework[39,40]. This broader perspective emphasizes the interconnectedness of pest management practices with ecological sustainability.

## Conclusion
The development and implementation of green crop protection is paramount for achieving various goals outlined in the 2030 agenda of the United Nations' Sustainable Development Goals (SDGs)[41], including food security and climate action. The 3MP theoretical framework offers valuable insights into more effective and sustainable management of multiple pests. It anticipates an increased adoption of green IPM tactics in both protected and open-field agriculture settings. This framework represents a crucial stride towards unlocking the full potential of IPM, ensuring food security, and simultaneously minimizing agriculture's global footprint. It aligns with the broader objectives of sustainable development encapsulated in the SDGs.

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

## Acknowledgements

We thank Dr. George Hamilton for comments on an earlier version of the manuscript. The study is funded by the project "National Key R&D Program of China" (2023YFE0104800)", the project from Yunnan Province Science and Technology Department "Yunnan International Joint Laboratory of Fruit-Vegetable-Flower Invasive Insect Pest Management (Yunnan FVF-IPM Joint Lab)" (No. 202303AP140018), and the project "Yunnan Xingdian Talent Support Program" (No. K264202230209).

## Author contributions

P.H. and N.D. conceptualized the study; P.H. wrote the original draft; P.H., C.R-S., M.Z., S-S.L. and N.D. reviewed and edited the manuscript.

## Competing interests

The authors declare no competing interests. N.D. is an Editorial Board Member for Communications Biology, but was not involved in the editorial review of, nor the decision to publish this article.
