## [Peer review file · Communications Biology]

Reviewers' comments:

Reviewer #1 (Remarks to the Author):

Communications Biology- Perspective

A theoretical framework toward a more effective 'green' crop protection
Peng Han, Cesar Rodriguez-Saona, Myron P. Zalucki, Shu-sheng Liu, Nicolas Desneux

Summary: This perspective paper promotes a new theoretical IPM framework, the Multi-dimensional Management of Multiple Pest (3MP), in an effort to improve the usage of green crop protection practices. They discuss their framework in relation to two key issues with the more commonly-used simple-mixed input approach: the issue of low synergy and low coverage across pest management tactics. The authors promote the use of more holistic methods that make green approaches more cost effective for growers.

General Comments: This article touches on some very important and relevant issues related to pest management and its environmental considerations, including grower hesitancy to utilize more green approaches and the short-comings of current IPM strategies (capitalizing on synergistic and multiple-pest approaches). Because of this, I do believe the topic is appropriate for the journal. That being said, the treatment of the subject is too superficial. Much more information about 'green' crop protection options and their advantages/disadvantages (with supporting data) is needed. In addition, the framework proposed by the authors is also very vague. It is clear that they are advocating for an IPM framework that considers multi-trophic interactions over time, but it isn't clear how this framework helps to actually incorporate this information into an IPM program (aside from discussing research suggestions). To me, the proposal of a framework requires much more criteria than is presented here. It currently reads like a list of suggestions for how to improve our understanding of pest dynamics in agricultural systems, which is useful and perfectly fine for a perspective article, but this does not constitute a "framework" in my opinion. If the authors' suggestions can be used to create clearer objectives for future research or IPM tactics, it will represent a better framework. Being 7 pages, there is a lot of room for the authors to elaborate on their ideas in more detail. The number of references is also too low for an adequate treatment of this topic. I would suggest starting with a more thorough review of current green tactics and their pros and cons, then discussing suggestions to increase grower usage.

Specific comments:

There is considerable usage of specific terms throughout the paper, such as SMI, green, and SDGs, yet often these are not clearly defined. I would suggest starting out by discussing what 'green' crop protection is in more detail. This could easily be a section on it's own, tying in the UN's SDGs and other climate change considerations. Doing this is particularly important because IPM already attempts to minimize environmental risk, but the reader needs to know what you consider 'green' tactics and how they fit into current IPM strategies. I would also ask whether the focus of your framework is on increasing the implementation of "green" tactics in IPM or producing a better version of IPM. Many of your recommendations focus on how information is implemented in an IPM program, rather than on promoting specific greener tactics through IPM, i.e., the consideration of bottom-up AND top-down effects of a given tactic.

Title: I would suggest a little more specific of a title, "A theoretical framework to improve the adoption of 'green' IPM tactics"

*Some wording and grammatical edits were made and can be found in the revised document.

Reviewer #2 (Remarks to the Author):

This manuscript proposes a new approach to integrated pest management: "Multi-dimensional Management of Multiple Pests" (3MP). This approach is meant to integrate today's updated understanding of our understanding of agroecology, which has advanced substantially since IPM

was coined decades ago. Top-down, bottom-up, multitrophic interactions, and many other aspects of our modern understanding of how pests, beneficials, and crops interact are discussed in the context of how pest management might be improved from a sustainability perspective, especially in the context of climate change.

The paper is well written. I have very few specific editorial remarks (see below). However, my main concern is that, in its current form, I do not find the authors' argument to be compelling. Essentially, they are advocating for integrated pest management, which already exists as a concept. How is 3MP significantly different from IPM? The main differences I see discussed are how pest management relates to climate change and environmental sustainability. These concepts have been included in discussions of IPM strategy/rationale and research for many years. What is the purpose of differentiating this approach from IPM to begin with? If there is not a significant payoff from a research or adoption perspective, I wonder if "rebranding" this would actually work against decades of work to get growers onboard with IPM. As it stands, getting growers to adopt evidence-based IPM practices can be a major struggle. Changing the name of it will not help, especially if it is primarily theoretical in nature.

Another aspect that the authors should develop more is what the cost-benefit of these tactics are for growers. If there is a clear economic or environmental benefit, stakeholders will be more likely to adopt these practices and concepts.

I think there is merit to what the authors discuss, but that the evidence and argument need to be reframed. I recommend that the authors address my comments above, and add specific examples and references throughout. As it stands, I am not convinced of the benefit of differentiating 3MP from IPM.

Major comments:
see above

Minor comments:

1. Line 26: change "nowadays" to "today"
2. Lines 79-80: reference needed
3. Line 87: change "biocontrol plants" to "plants that support biological control"
4. Line 89: change "lure" to "attract"
5. Lines 87-89: reference needed
6. Lines 103-104: I think that there is a growing body of literature on this. I suggest framing that as more of a historical issue than a current one
7. Lines 132-135: I think is mostly hypothetical. Often these changes have major tradeoffs (e.g. spraying organic materials often means more applications compared to conventional). It is hard to know what the net cost or benefit might be. This section should be expanded.

Dear reviewers,

Thanks a lot for your useful comments, which enables us to have a much more informative perspective. Below are our point-by-point responses to your comments. Accordingly, we show where and how we made changes in the “Revised Manuscript with highlighted changes” file.

Reviewers’ comments:

Reviewer #1 (Remarks to the Author):

Communications Biology- Perspective

A theoretical framework toward a more effective ‘green’ crop protection
Peng Han, Cesar Rodriguez-Saona, Myron P. Zalucki, Shu-sheng Liu, Nicolas Desneux

Summary: This perspective paper promotes a new theoretical IPM framework, the Multi-dimensional Management of Multiple Pest (3MP), in an effort to improve the usage of green crop protection practices. They discuss their framework in relation to two key issues with the more commonly-used simple-mixed input approach: the issue of low synergy and low coverage across pest management tactics. The authors promote the use of more holistic methods that make green approaches more cost effective for growers.

General Comments: This article touches on some very important and relevant issues related to pest management and its environmental considerations, including grower hesitancy to utilize more green approaches and the short-comings of current IPM strategies (capitalizing on synergistic and multiple-pest approaches). Because of this, I do believe the topic is appropriate for the journal.

Thank you for the positive comments.

That being said, the treatment of the subject is too superficial. Much more information about ‘green’ crop protection options and their advantages/disadvantages (with supporting data) is needed.

We agree in part. Our perspective paper is not focusing on the advantages/disadvantages of individual crop protection options (we do focus on the “low integration level” when the management options are simultaneously adopted by growers, so-called Simple Mixed Inputs (SMI) approach.), we have added some more information in the first paragraph. Those added lines could help readers to get a better transition to the following paragraphs. Adding supporting data we believe would move beyond a “perspective” into a major review.

Inspired by your comments, we have provided some more specific cases to show how

“low synergy” and “low coverage” usually look like in IPM programs (see the revised Paragraph 3 and 4). This could help readers to understand better what are the bottlenecks.

In addition, the framework proposed by the authors is also very vague. It is clear that they are advocating for an IPM framework that considers multi-trophic interactions over time, but it isn't clear how this framework helps to actually incorporate this information into an IPM program (aside from discussing research suggestions). To me, the proposal of a framework requires much more criteria than is presented here. It currently reads like a list of suggestions for how to improve our understanding of pest dynamics in agricultural systems, which is useful and perfectly fine for a perspective article, but this does not constitute a “framework” in my opinion. If the authors' suggestions can be used to create clearer objectives for future research or IPM tactics, it will represent a better framework.

Point taken. IPM paradigm encompasses various aspects including knowledge & resources, planning & organization, communication, and pest management (as shown by Dara 2019). Obviously, we are mainly touching the last aspect - how the adoption of various management options could be optimized to meet the food security while minimize environmental impacts. Indeed, many of the IPM ‘green’ tools and technologies do not reach the implementation stage in fields owing to low cost-effectiveness, which is largely ascribed to the lack of knowledge on the interactions between/among those tactics by growers and even the outreach specialists and researchers. Our framework aims to help researchers to fill this knowledge gap, and help outreach specialists design “greener” versions of IPM packages in a specific crop system.

Dara S.K. (2019) The New Integrated Pest Management Paradigm for the Modern Age. *Journal of Integrated Pest Management*. 12:1-9

We have undertaken revisions on two aspects:

(1) To have a clearer objective of the framework, we have provided more details in the paragraph 2, the section “New Theoretical Framework”, and the section “A Holistic Approach: could it help us design greener IPM packages?”.

(2) To have clearer criteria for assessing the benefits of relying on 3MP, we have added one separate section just before the conclusion section - “Criteria for assessing the increased adoption of ‘green’ tactics in IPM”.

Being 7 pages, there is a lot of room for the authors to elaborate on their ideas in more detail. The number of references is also too low for an adequate treatment of this topic. I would suggest starting with a more thorough review of current green tactics and their pros and cons, then discussing suggestions to increase

grower usage.

We have enriched this part with more details (see Paragraph 1) as well as in the rest sections. In the current version, we have sufficient references for elaborating on our ideas in more detail.

Specific comments:

There is considerable usage of specific terms throughout the paper, such as SMI, green, and SDGs, yet often these are not clearly defined. I would suggest starting out by discussing what ‘green’ crop protection is in more detail. This could easily be a section on it’s own, tying in the UN’ s SDGs and other climate change considerations. Doing this is particularly important because IPM already attempts to minimize environmental risk, but the reader needs to know what you consider ‘green’ tactics and how they fit into current IPM strategies.

DONE. We have discussed what various ‘green’ management tactics are in more detail in the first paragraph. We have also enriched the description of those specific terms. Citations of various types of ‘green’ tactics in general have been added.

I would also ask whether the focus of your framework is on increasing the implementation of “green” tactics in IPM or producing a better version of IPM. Many of your recommendations focus on how information is implemented in an IPM program, rather than on promoting specific greener tactics through IPM, i.e., the consideration of bottom-up AND top-down effects of a given tactic.

Very insightful comment. The focus of our framework is on the former – increasing the implementation of “green” tactics in IPM (i.e. ‘greener’ IPM packages). The reviewer 2 also raised this important issue. We believe it is now more clearly stated (see the Paragraph 5 and the section “New Theoretical Framework”).

Title: I would suggest a little more specific of a title, “A theoretical framework to improve the adoption of ‘green’ IPM tactics”

Fully agree. We have used this new title.

*Some wording and grammatical edits were made and can be found in the revised document.

Done.

Reviewer #2 (Remarks to the Author):

This manuscript proposes a new approach to integrated pest management: “Multi-dimensional Management of Multiple Pests” (3MP). This approach is meant to integrate today’ s updated understanding of our understanding of agroecology, which has advanced substantially since IPM was coined decades ago. Top-down, bottom-up, multitrophic interactions, and many other aspects of our modern

understanding of how pests, beneficials, and crops interact are discussed in the context of how pest management might be improved from a sustainability perspective, especially in the context of climate change.

The paper is well written. I have very few specific editorial remarks (see below). However, my main concern is that, in its current form, I do not find the authors' argument to be compelling. Essentially, they are advocating for integrated pest management, which already exists as a concept. How is 3MP significantly different from IPM? The main differences I see discussed are how pest management relates to climate change and environmental sustainability. These concepts have been included in discussions of IPM strategy/rationale and research for many years. What is the purpose of differentiating this approach from IPM to begin with? If there is not a significant payoff from a research or adoption perspective, I wonder if "rebranding" this would actually work against decades of work to get growers onboard with IPM. As it stands, getting growers to adopt evidence-based IPM practices can be a major struggle. Changing the name of it will not help, especially if it is primarily theoretical in nature.

Thanks for pointing out the concern (also raised by the reviewer 1), which indeed helps to re-think our framework. As you mentioned, we do advocate IPM, which has been a key paradigm guiding pest management for decades. We do not intend to create a new strategy against the principles of IPM (nor differentiate it from IPM). Instead, the focus of our framework is aiming to increase the implementation/adoption of green tactics (i.e., the integration level of various management tactics) in IPM where the issue of low synergy and low coverage of pest management tactics have been overlooked. This has been one of the major bottlenecks limiting higher adoption of IPM by growers (a major struggle for them as you have mentioned), especially the 'green' tactics.

Our 3MP framework is supposed to address this key issue. It will pay off by offering a perspective encouraging researchers and end-users to rely on the holistic approach to think and design IPM packages with increased synergy and coverage of IPM tactics. Overall, 3MP does not intend to rebrand IPM, it is a framework which may lead to a higher adoption of 'green' IPM tactics because of being more cost effective for farmers and deeper consideration on environmental well-beings.

We have added more lines in the revised manuscript (see the Paragraph 5 and the section "New Theoretical Framework").

Another aspect that the authors should develop more is what the cost-benefit of these tactics are for growers. If there is a clear economic or environmental benefit, stakeholders will be more likely to adopt these practices and concepts. We do agree with you. This issue has also been mentioned by reviewer 1. We have added one separate paragraph to address this issue (just before the

conclusion paragraph).

I think there is merit to what the authors discuss, but that the evidence and argument need to be reframed. I recommend that the authors address my comments above, and add specific examples and references throughout. As it stands, I am not convinced of the benefit of differentiating 3MP from IPM.

Done.

Major comments:

see above

Minor comments:

1. Line 26: change “nowadays” to “today”

Done.

2. Lines 79–80: reference needed

Done.

3. Line 87: change “biocontrol plants” to “plants that support biological control”

Done. We agree that it is misleading to use “biocontrol plants”, we dig more the literature (Jaworski et al. 2023), and we decide to use “functional plants that support biological control”. It is also updated in the figure.

4. Line 89: change “lure” to “attract”

Done.

5. Lines 87–89: reference needed

Done. We have cited a very recent review relevant to this topic (Jaworski et al. 2023).

6. Lines 103–104: I think that there is a growing body of literature on this. I suggest framing that as more of a historical issue than a current one

Done. Indeed, there is growing body of literature on the indirect interactions (It has been reviewed by E. Poelman and M. Dicke in 2014, and Emery and Mills 2020). Herein, what we want to convey is that IPM specialists seldom consider those indirect interactions when designing and making IPM packages practical in fields. Thus, to avoid misleading message, we have revised this sentence. It now reads “Despite a growing body of literature has shown those indirect interactions (Emery and Mills 2020; Poelman and Dicke, 2014), unfortunately, they are often ignored during crop protection practice.”

7. Lines 132–135: I think is mostly hypothetical. Often these changes have major tradeoffs (e. g. spraying organic materials often means more applications compared

to conventional). It is hard to know what the net cost or benefit might be. This section should be expanded.

Done. We do agree with your point. To avoid the hypothetical statements, we have revised this sentence with a more cautious tone.

In addition, we have made changes to the figure. It is possible that the bottom-up forces could reach the third and higher trophic levels, but more often, the effects are unknown and might be case-specific. The figure legend has also been updated accordingly.

The format of the references citation and list are now fitting the standard of the journal.

We look forward to your further feedback.

Sincerely yours,

Peng Han & Cesar Rodriguez-Saona

REVIEWERS' COMMENTS:

Reviewer #1 (Remarks to the Author):

As mentioned in my last review of the manuscript, this article touches on some very important and relevant issues related to pest management and its environmental considerations, including grower hesitancy to utilize more green approaches and the short-comings of current IPM strategies (capitalizing on synergistic and multiple-pest approaches). As such, this perspective paper is very appropriate for the journal and discusses important and timely topics in the field of ecology and agricultural sciences.

I feel that the authors did a great job in considering my recommendations and making changes to the manuscript to address my concerns and/or suggestions. In particular, they added important information to better describe some key concepts and more clearly outlined their framework. I feel that the manuscript is much improved and more clearly describes authors points. I am very satisfied with their edits.

Reviewer #3 (Remarks to the Author):

With respect to the manuscript "A theoretical framework to improve the adoption of 'green' IPM tactics", examining both the document, the comments from the reviewers, and responses, I am of a mixed opinion. The authors touch on some very important and relevant concerns and thoughts. I cannot agree more that aspects such as trophic levels and seasonality, in addition to 'green' methods should be considered. Although, as with the second reviewer I am not convinced this is not simple a component of modern IPM.

It is quite clear that the authors took substantial efforts to address the reviewers' comments and concerns. Most of the more factual and contextual issues have been addressed. I am, however, a little unsure about the more substantial concerns about the aim of the piece. There seems to be some disconnect between the goal to encourage "green" methods, the concept of 3MP introduced by the authors, and the trophic and temporal aspects touched upon. As with the first reviewer I am a little lost as to whether this is focused on the green aspect, in which there is still much room for elaboration, or if the green aspect is a component (an important one) of the proposed 3MP approach. Perhaps this all reduces to a need for the authors to provide a better and more precise explanation of the approach. I think that is what they are attempting in lines 115-125 but, in my opinion, they do not quite succeed. It might be helpful to provide a table that shows how their approach directly compares and contrasts with existing IPM and maybe IPPM approaches.

Dear editor and reviewers:

We thank for your further comments on the draft. We reply to your comments point by point as below and the changes are made in the manuscript.

REVIEWERS' COMMENTS:

Reviewer #1 (Remarks to the Author):

As mentioned in my last review of the manuscript, this article touches on some very important and relevant issues related to pest management and its environmental considerations, including grower hesitancy to utilize more green approaches and the short-comings of current IPM strategies (capitalizing on synergistic and multiple-pest approaches). As such, this perspective paper is very appropriate for the journal and discusses important and timely topics in the field of ecology and agricultural sciences.

I feel that the authors did a great job in considering my recommendations and making changes to the manuscript to address my concerns and/or suggestions. In particular, they added important information to better describe some key concepts and more clearly outlined their framework. I feel that the manuscript is much improved and more clearly describes authors points. I am very satisfied with their edits.

Thanks a lot for your earlier thoughtful comments, which are indeed very helpful. Thanks again!

Reviewer #3 (Remarks to the Author):

With respect to the manuscript “A theoretical framework to improve the adoption of ‘green’ IPM tactics”, examining both the document, the comments from the reviewers, and responses, I am of a mixed opinion. The authors touch on some very important and relevant concerns and thoughts. I cannot agree more that aspects such as trophic levels and seasonality, in addition to ‘green’ methods should be considered. Although, as with the second reviewer I am not convinced this is not simple a component of modern IPM.

It is quite clear that the authors took substantial efforts to address the reviewers’ comments and concerns. Most of the more factual and contextual issues have been addressed. I am, however, a little unsure about the more substantial concerns about the aim of the piece. There seems to be some disconnect between the goal to encourage “green” methods, the concept of 3MP introduced by the authors, and the trophic and temporal aspects touched upon. As with the first reviewer I am a little lost as to whether this is focused on the green aspect, in which there is still much room for elaboration, or if the green aspect is a component (an important one) of the proposed 3MP approach. Perhaps this all

reduces to a need for the authors to provide a better and more precise explanation of the approach. I think that is what they are attempting in lines 115–125 but, in my opinion, they do not quite succeed. It might be helpful to provide a table that shows how their approach directly compares and contrasts with existing IPM and maybe IPPM approaches.

Thank you for your comments on this piece of perspectives. We would like to emphasize that we do not focus on “green” tactics themselves, instead, what we focus on is how these tactics, as components of IPM, should be properly used based on our 3MP approach. We feel that it could be helpful if we can show how our 3MP fits with IPM. For this point, we have provided an additional figure (Figure 2) to show the information. One sentence has been added in the end of the section “New Theoretical Framework”. All the edits have been highlighted in yellow in the file.

We hope that these revisions address this concern, and look forward to your further comments.

Additionally, we realize that funding information should be provided. We have thus updated this information. The affiliation of one co-author has been updated based on his approval. During this revision, we have also edited our manuscript (notably the reference list) to comply with the format requirements of the journal. The two figures are also provided in a separate PPT file to facilitate further editorial process upon the confidential acceptance.

Best wishes

Peng Han (on behalf of all authors.)

penghan@ynu.edu.cn

Yunnan University